# Exploring unsupervised top tagging using Bayesian inference

Ezequiel Alvarez[1][*], Manuel Szewc[2][†], Alejandro Szynkman[3][‡], Santiago Tanco[3][§],
Tatiana Tarutina[4][¶]

[1]*International Center for Advanced Studies (ICAS) and CONICET, UNSAM,
Campus Miguelete, 25 de Mayo y Francia, CP1650, San Martín, Buenos Aires, Argentina*

[2]*Department of Physics, University of Cincinnati,
Cincinnati, Ohio 45221,USA*

[3]*IFLP, CONICET - Dpto. de Física, Universidad Nacional de La Plata,
C.C. 67, 1900 La Plata, Argentina*

[4]*IFLP, CONICET,
Diagonal 113 e/ 63 y 64, 1900 La Plata, Argentina*

## Abstract

Recognizing hadronically decaying top-quark jets in a sample of jets, or even its total fraction in the sample, is an important step in many LHC searches for Standard Model and Beyond Standard Model physics as well. Although there exists outstanding top-tagger algorithms, their construction and their expected performance rely on Montecarlo simulations, which may induce potential biases. For these reasons we develop two simple unsupervised top-tagger algorithms based on performing Bayesian inference on a mixture model. In one of them we use as the observed variable a new geometrically-based observable $\tilde{A}_3$, and in the other we consider the more traditional $\tau_3/\tau_2$ $N$-subjettiness ratio, which yields a better performance. As expected, we find that the unsupervised tagger performance is below existing supervised taggers, reaching expected Area Under Curve AUC $\sim 0.80 - 0.81$ and accuracies of about $69\% - 75\%$ in a full range of sample purity. However, these performances are more robust to possible biases in the Montecarlo that their supervised counterparts. Our findings are a step towards exploring and considering simpler and unbiased taggers.

[*]sequi@unsam.edu.ar
[†]szewcml@ucmail.uc.edu
[‡]szynkman@fisica.unlp.edu.ar
[§]santiago.tanco@fisica.unlp.edu.ar
[¶]tarutina@fisica.unlp.edu.ar

# 1  Introduction

After an outstanding successful first half of the LHC lifespan, we are heading for a second half in which one of the most important features is data gathering. Consequently, one of the main concerns for the community is how to take best and full advantage of a unique dataset of protons colliding in a controlled environment. Among the most relevant studies for this stage are those including heavy particles such as the top quark, the Higgs boson and the heavy gauge bosons $Z$ and $W$, since they are not only important by themselves, but also because there are theoretical arguments to expect that these are the Standard Model (SM) particles with larger couplings to New Physics (NP). Improving identification techniques for these particles in a variety of scenarios at the LHC is a crucial tool for the search of NP. Along this work we study innovative techniques for identifying boosted top quarks within a framework of Bayesian unsupervised Machine Learning.

The motivations for pursuing the study of tops are multiple. The top quark is the heaviest particle of the SM and plays crucial roles both in the consistency of the model through the electroweak precision tests and in the stability of the electroweak vacuum. Tops might also have a special participation in the mechanism of electroweak symmetry being a particularly interesting window to NP. Moreover, due to its short life time, the top quark decays before that hadronization takes place and this leads to the only scenario where it is possible to analyze the physics of a unbound quark. The top quark was discovered at Tevatron in 1995 [1,2], and since then it has become an object of exhaustive analysis [3]. With the start of the LHC, which can be seen as a top-factory, the study of the top quark entered into a new stage [4–9] with unique challenges. In particular, because of the high energies achieved at the LHC, highly boosted top quarks with transverse momentum above $p_T \gtrsim 300$ GeV are copiously produced. In this kinematic regime, the top decay products become collimated and the top jet is reconstructed as a fat jet by the standard jet clustering algorithms. Consequently, the techniques for identifying top quarks change drastically with respect to the resolved regime, relying on the analysis of the jet substructure. Following the proposal of building taggers based on jet substructure introduced in Ref. [10], the study of top jet substructure for top quark identification has developed into an area of collider physics coined as *top-tagging*, which has reached many milestones [11].

The main challenge within this area is to distinguish hadronically decaying top quarks collimated as fat jets, which we shall call *top jets*, and jets that are induced by quarks and gluons, which we will name hereafter as *QCD jets*. This is of course a very difficult task, since it implies –among others– understanding and/or modelling hadronization, splittings and showering processes which occur at an energy level in which QCD is not totally understood. Traditional top taggers combine standard clustering algorithms, jet grooming techniques, declustering, mass drop, and soft drop among others, in order to search for kinematic features in the jet [12–19]. With the emergence of high performance Machine Learning algorithms and associated boost in available hardware, even more powerful techniques have been developed [20–31], usually relying on Neural Networks trained to distinguish top from light quark and gluon QCD jets using as a training set labelled Montecarlo generated samples. In the last few years, these techniques have improved considerably by defining and using specific objects within jets, such as for instance the Energy Flow Polynomials [28]. The state of the art in these techniques reach an oustanding tagging performance which is measured as an area under the ROC curve, AUC $\approx$ 0.988, and an accuracy of about 94% [32, 33] using the standard Top Quark Tagging Reference Dataset [34]. However because these are supervised techniques, the tagger performance is contingent on the specific Montecarlo simulations considered for labelled training and benchmarking at a given working point within the ROC curve. Any potential bias in the Montecarlo would lead to a bias in the use of the algorithm, and thus in any conclusions drawn from it; see for instance recent discussion in [35,36]. A clear example of possible issues in top physics arising due to Montecarlo simulations is described in Sec. 3.2. of Ref. [37]. In general, obtaining precise Montecarlo predictions for differential distributions can be challenging due to the need to model color reconnection, Multiparticle Interactions and hadronization effects, while the definition of non-zero widths and even of the perturbative and non-perturbative top quark mass parameters in Montecarlo can lead to uncertainties. The effect of these modelling assumptions on predictions are usually determined by a combination of parameter scan and different choices of Montecarlo simulators, but the uncertainty determination and necessary template morphing of the obtained uncertainties is an arbitrary choice which could prove to underestimate the true uncertainties. Because of this, it is worth exploring other methods, such as unsupervised top-tagging techniques, even though they may be below the performances announced within supervised frameworks. An unsupervised approach, such as Latent Dirichlet Allocation (LDA) [30], could be run directly over experimental data, bypassing potential simulation biases, but generally giving a less efficient tagging performance.

In this manuscript we focus on exploring other unsupervised alternatives, by means of Bayesian inference over a mixture model [38] to simultaneously learn about top and QCD jet features, estimate the mixture fractions in a given sample, and create a top-tagging algorithm. Bayesian techniques are based in designing a Probability Density Function (PDF) from which one assumes that the data is sampled, and then, using the measured data, a probability distribution for the parameters of said PDF is inferred. The advantage comes from using robust knowledge to construct the PDF and leaving the unknowns as random variables whose distribution is to be inferred. A mixture model considers the particular case where each jet belongs to a given class, which in this manuscript are either top or QCD jets. Mixture models have previously been applied in High Energy Physics for samples containing signal and background [39–41]. In designing the PDF that can distinguish top jets from QCD jets, one should use robust knowledge. The use of mixed membership models such as LDA and mixture models for LHC physics is part of a surge of generative modelling of LHC phenomenology. However, they are of a different nature than most current implementations of generative networks which are based on variations of Autoencoders (see e.g. [42–46]), Generative Adversarial Networks (see e.g. [44, 47]), Normalizing Flows (see

e.g. [46, 48–50]), Conditional Invertible Neural Networks (see e.g. [44, 48, 51]) or even Diffusion models [52]. These usually either build surrogate models for simulators, which we wish to ignore in this work, or cannot accommodate the multi-class nature of the underlying probability distribution. The most noticeable exception is Ref. [43] where the authors explore different latent space prior possibilities, one of them being a gaussian mixture model. However, in that work they are not able to match one-to-one the learned mixtures with the physical underlying processes. We aim to learn simple probability densities with as little dependence on simulators as possible and where we can match the underlying themes with the physical processes at the expense of loosing discrimination power.

Along this manuscript we exploit the fact that the top has a well known hadronic decay to three particles, $t \to Wb \to jjb$. We work within two different frameworks, one of which appeals to a simple geometrical representation of the process whereas the other employs $N$-subjettiness [53] observables. Implementing these techniques in such a way that the robust knowledge is used to determine the observable and the shape of its PDF, while the unknown parameters left to be inferred from the data, is part of the *art et métiers* of a Bayesian inference framework, and it is what is pursued in the next Sections.

This work is divided as follows. In the next Section we discuss Bayesian inference techniques on mixture models for tagging at the LHC. In Sec. 3 we perform inference on a mixture model in which the observed variable is the third coefficient of the Fourier transform of the distribution of tracks around the center of the jet. In Sec. 4 we infer on a mixture model which takes as observed variable the well-known $N$-subjettiness ratio $\tau_3/\tau_2$. In Sec. 5 we discuss the obtained results and we summarize the conclusions of the article.

## 2 Mixture models for top-tagging at the LHC

In this Section we provide a brief summary of mixture models oriented to top-tagging. For each jet, we define a tagging observable $x$ whose distribution we propose as a mixture model, in which the likelihood function for a collection $X$ of $N$ independent measurements of this observable can be expressed as

$$p(X) = \prod_{i=1}^{N} p(x_i) = \prod_{i=1}^{N} \sum_{k=\{0,1\}} \pi_k \, p(x_i|k) \,, \tag{1}$$

where $\pi_k$ is the probability of sampling a jet from jet class $k$ and $p(x|k)$ the class-dependent probability functions for the tagging observable $x$. We use $k = 0\,(1)$ to denote QCD (top) jets. The mixture model is defined by providing appropriate probability densities $p(x|k)$ which need to capture the relevant physics. These probability densities are typically parametric functions dependent on a set of a few latent parameters $\zeta_k$. For a given mixture model one can find the set of parameters $\boldsymbol{\zeta} = \{(\pi_k, \boldsymbol{\zeta}_k), k = 0, 1\}$ that most accurately describe the observed data distribution in typical frequentist fashion by maximizing the Likelihood introduced in Eq. 1. In this work, we consider a Bayesian framework where the parameters are promoted to random variables and it is possible to obtain their posterior probability distribution $p(\boldsymbol{\zeta}|X)$ by first proposing a prior probability $p(\boldsymbol{\zeta})$ and using Bayes' theorem,

$$p(\boldsymbol{\zeta}|X) \propto p(X|\boldsymbol{\zeta})p(\boldsymbol{\zeta}) \,, \tag{2}$$

where $p(X|\boldsymbol{\zeta})$ is given by Eq. (1). The posterior distribution can be estimated via approximate Bayesian inference techniques. Alternatively, there are techniques to directly obtain the set of model parameters that maximize the posterior probability, i.e. the maximum a posteriori (MAP). Along this work we use the latter option through Stochastic Variational Inference (SVI) [54], via the `Pyro` package [55] in `Python3`.

In Variational Inference (VI), the posterior $p(\boldsymbol{\zeta}, z|X)$ is approximated through a function family $q(\boldsymbol{\zeta}, z)$, where $z$ are the hidden local variables such as class assignment. The specific function of the family $q$ is chosen by maximizing the Evidence Lower Bound (ELBO) between the posterior proxy $q$ and the joint distribution $p(X, \boldsymbol{\zeta}, z)$,

$$\mathbb{E}_q[\log p(X, \boldsymbol{\zeta}, z)] - \mathbb{E}_q[\log q(\boldsymbol{\zeta}, z)] = \int q(\boldsymbol{\zeta}, z) \left(\log p(X, \boldsymbol{\zeta}, z) - \log q(\boldsymbol{\zeta}, z)\right) d\boldsymbol{\zeta} dz \qquad (3)$$

which is equivalent to minimizing the Kullback-Leibler divergence between the true posterior and its approximation, capturing how much information is lost when approximating the distribution $p(\boldsymbol{\zeta}, z|X)$ with the distribution $q(\boldsymbol{\zeta}, z)$. The key of the method is to choose an adequate function family $q$ that renders the ELBO tractable. That is, a function family for which the expectation values in the ELBO are efficiently computed. The simplest choice for full posterior distribution is the mean field approximation. To estimate the MAP using SVI the choice of $q$ is even simpler: we propose as a posterior a product of Dirac delta distributions for the $\boldsymbol{\zeta}$, one for each parameter.

$$q(\boldsymbol{\zeta}, z) = \delta(\boldsymbol{\zeta} - \boldsymbol{\zeta}^{\text{MAP}})q(z|\boldsymbol{\zeta}) \qquad (4)$$

where $q(z|\boldsymbol{\zeta}^{\text{MAP}})$ is the conditional local class asignment probability and $\boldsymbol{\zeta}^{\text{MAP}}$ the desired MAP values. By substituting the appropriate joint distribution on the ELBO and computing the gradients, it can be shown that $q(z|\boldsymbol{\zeta}^{\text{MAP}})$ will be a Categorical distribution for each event with parameters depending on $\boldsymbol{\zeta}^{\text{MAP}}$ and $x$. The multinomial parameters and more importantly the $\boldsymbol{\zeta}^{\text{MAP}}$ can be computed by gradient descent obtaining equivalent results to standard Expectation-Maximization with a prior.

As usual with large datasets, performing Gradient Descent can be very memory consuming. To avoid this, SVI introduces noisy gradient estimators by computing the gradient with subsets of data. The parameters are then estimated locally for each data subsample and updated globally with an additional hyperparameter to control the learning rate of the stochastic procedure.

In this work, we will thus learn $\boldsymbol{\zeta}^{\text{MAP}}$ from unlabelled data and build a top tagger using the learned mixture model. Although in this manner we avoid relying on Montecarlo simulations, there will still be sources of bias. In particular, the appropriateness of $p(x|k)$ will determine the bias of the model. Because the unsupervised framework aims to reproduce the total density, our model may be successful without capturing the true underlying distributions, a problem which worsens the more the truth distributions divert from the chosen parametric families. There are different methods to examine and evaluate this kind of bias [56], in this work we choose to compute an adaptation to the MAP of a posterior predictive check. This check is performed on the total distribution with the unlabelled data.

The posterior predictive check consists in quantifying how well the parameters of the model extracted in the inference procedure can generate replicates of the data which is statistically similar to the true data. To perform this check, one can sample model parameters from the posterior, and use them to sample replicates of the data, $X_{rep}$. The check comes from studying how statistically

similar are the replicate data set and the true data set from the statistic point of view. There is not a unique nor standard way of doing this, along this work we perform this assessment following Ref. [57], and with the adaptation that instead of sampling from the posterior, we use the MAP parameters.

Given the true data set, a part of it is held out ($X_{held}$) and it is not seen by the inference procedure. This division is usually referred as training and test data sets. We use the training data set to extract the MAP using Bayesian inference, and with the MAP parameters we have the PDF of the $X_{rep}$ that would be sampled from it[1]. We call this the replicate PDF. Following Ref. [57] we define a predictive score as the probability that the probability of sampling the held-out data set with the replicate PDF, would have been smaller than its actual value:

$$\text{predictive score} = p\big( p(X_{held}) < p(X_{rep}) \big). \tag{5}$$

To obtain the predictive score, one can simply compute the area under the replicate PDF in which the probability is less than $p(X_{held})$. If the true data would have been sampled with the replicate PDF, then one should expect a predictive score of 0.5. Scores above 0.5 would mean that the model is misspecified: although the model is correctly predicting where the data is, it is also predicting data in a broader region where it is not. Scores below 0.5 would indicate that the model is predicting a replicate PDF that is biased with respect to the true data. Depending on the level of information available one may use different (subjective) thresholds to consider a model suitable. As a reference, in [57], where the model should reproduce a matrix from social data as a product of two matrices with considerable less dimensions, they use a predictive score threshold of 0.10 to consider a model suitable. In our case, the data distribution comes from a physical system and hence we can expect higher (subjective) thresholds.

Along this work, in addition to the above posterior predictive check on the full distribution, we also perform a similar analysis on the labelled data to examine the goodness of the model. This is only possible in synthetic data, where we can access the labels.

In order to construct a top tagger from the Bayesian inference process described above, one can get the MAP using SVI, and compute the assignment probability

$$p(\text{QCD}|x, \boldsymbol{\zeta}^{\text{MAP}}) = \frac{\pi_0^{\text{MAP}} p(x|\boldsymbol{\zeta}_0^{\text{MAP}})}{\sum_{k=\{0,1\}} \pi_k^{\text{MAP}} p(x|\boldsymbol{\zeta}_k^{\text{MAP}})} , \tag{6}$$

with $p(\text{top}|x, \boldsymbol{\zeta}^{\text{MAP}}) = 1 - p(\text{QCD}|x, \boldsymbol{\zeta}^{\text{MAP}})$. A given jet with an observed value of $x$ is then tagged as a QCD jet if its assignment probability is higher than a selected threshold, $p(\text{QCD}|x, \boldsymbol{\zeta}^{\text{MAP}}) > t$ with $0 < t < 1$.

In the following Sections we explore two choices for the tagging observable $x$ and its associated proposed mixture model. In Sec. 3 we introduce a new variable based on the 3-pronged nature of top jets and defined on the Fourier space. In Sec. 4 we use the ratio of $N$-subjettiness variables $\tau_3/\tau_2$, which shows a good discriminating power for tops.

Although we design an unsupervised algorithm that aims to be applicable directly to data, we make use of simulations to benchmark our proposal. To compare with other approaches more directly, we use the standard Top Quark Tagging Reference Dataset [34], which consists of 1.2M training events, 400k testing events and 400k validation events generated via `Pythia8` [58] at

---

[1] Observe that since we deal with the MAP instead of the posterior, we do not need to sample $X_{rep}$ to obtain its PDF. We use directly the MAP parameters to obtain $p(X_{rep})$.

c.m. energy of 14 TeV and with the default `Delphes` [59] simulation of the ATLAS detector response. Fat jets are reconstructed with the anti-$k_T$ algorithm at $R = 0.8$ and are required to have $p_T \in (550, 650)$ GeV [2] and $|\eta| < 2$. We work with the train dataset, first applying a selection cut in the invariant mass of the jets $m_j$, such that $145$ GeV $< m_j < 205$ GeV. Posterior predictive checks are performed on the test dataset after applying the same selections cuts.

Using this dataset, we always consider three different metrics to evaluate the learned model: the learned fraction to be compared against the true fraction, the accuracy when tagging at the fixed probabilistic threshold $t = 0.5$ and the AUC. The last two are standard in the top-tagging community. However, we emphasize that our method is not meant to maximize these supervised metrics through a smart choice of observables and algorithms. We only need our choice of observables and density modelling to perform well enough so that overcoming the biases of simulations is a clear benefit without a large cost in tagging performance or the new source of bias from the choice of $p(x|k)$ becoming more impactful than the simulator uncertainties. One should keep in mind that the bias from the model is more easily verified that the impact of possible biases in the Montecarlo simulation and that these checks are part of the Bayesian model-bulding framework.

## 3   Mixture model in the Fourier framework

We want to work with observables that take advantage of the well understood physical differences between QCD and top jets. In particular, since a top jet can be interpreted as a superposition of three subjets, we first introduce a geometrical variable that captures this feature. In the $(\theta, \varphi)$ space of the detector, each jet consists of a collection of coordinate pairs for each constituent track

$$(\text{Jet})_i = \{(\theta^{(j)}, \varphi^{(j)})_i \,, \; j = 1, \ldots, N_i\} \,. \tag{7}$$

The constituent coordinates are all shifted in such a way that the center of momentum of the jet coincides with $(\theta, \varphi) = (0, 0)$. We define the angle $\varpi$ with respect to the $\theta$ positive axis, such that $\varpi \in (-\pi, \pi)$, and heretofore we reduce the jets in our dataset to a collection of $N_i$ angles $\{\varpi_1, \varpi_2, \ldots, \varpi_{N_i}\}$, each corresponding to a particular track. Observe that we do not use the distance of the tracks to the center of momentum of the jet. We now take 20 equally sized bins in this variable, which define angular slices of the 2-dimensional plane $(\theta, \varphi)$, and count the number of tracks in each bin, resulting in an array of $N_{\text{bins}}$ integer numbers,

$$(a_0, a_1, \ldots, a_{N_{\text{bins}}-1}) \,, \tag{8}$$

for each event. The aforementioned reduction aims to make explicit an arbitrary degree of freedom for the $(a_i)$ histogram given by a rotation of all the jet constituents by the same angle around the center. We eliminate this arbitrariness by evaluating a discrete Fourier transform (DFT),

$$A_l = \sum_{m=0}^{N_{\text{bins}}-1} a'_m \, e^{-2\pi i \frac{ml}{N_{\text{bins}}}} \,, \quad l = 1, ..., 10 \,, \tag{9}$$

---

[2] We have not explored bins with higher $p_T$ but we expect a degradation in performance to discriminate between top and QCD jets since they tend to have a more similar substructure as $p_T$ increases. In particular, this is the case for $N$-subjettiness where 3-subjettiness resembles 2-subjettiness as the overlap between three subjets is larger at high $p_T$ [53].

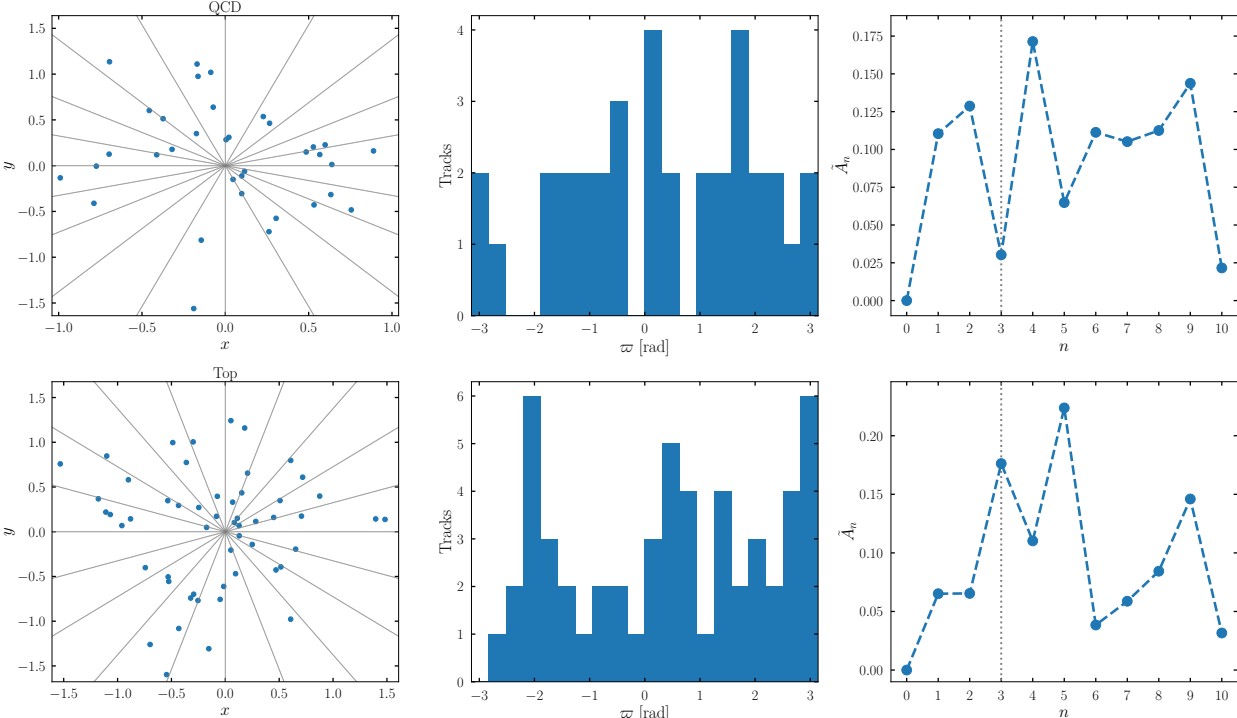

Figure 1: Left column: jet constituents in a bidimensional space of coordinates $(x, y)$, with angular slices shown by gray lines. Center column: histogram of constituents per angular slice. Right column: normalized values of the DFT results $\tilde{A}_n$, with $\tilde{A}_3$ highlighted by the dotted line. Top (bottom) row corresponds to a QCD (top) jet. Constituents were drawn from a toy model described in Appendix A.

where $a'_i = a_i - \bar{a}$, and $\bar{a}$ is the mean of $(a_i)$. Therefore, the baseline $A_0$ frequency is removed. Finally we normalize the $|A_l|$ values, by defining

$$\tilde{A}_l = \frac{|A_l|}{\sum_i |A_i|} \, . \tag{10}$$

As we expect the top jets to show a higher density of tracks around the centers of the three subjets, the DFT of the angular variable should show a tendency towards higher values in the $l = 3$ frequency of the top events with respect to the QCD background.

To illustrate this new set of variables we build a toy model, described in Appendix A, which also has two classes. These two classes reflect the expected differences between QCD and top jets, with the former generating random points in 2d space while the latter having a clear three-pronged structure. We show in Fig. 1 two typical examples of this pipeline for a toy model with a typical toy QCD (top) jet shown in the upper (lower) row. In the left column we show the constituents in the $(\theta, \varphi)$ plane, called $(x, y)$ for the toy model, with angular bins drawn in gray. In the middle column we have the histogram for the angular variable $\varpi$ that represents the $(a_i)$ values, and the corresponding $\tilde{A}_l$ distribution is shown in the right column. Fig. 1 shows that the resulting $\tilde{A}_n$, in particular $\tilde{A}_3$, may highlight the three-pronged structure of top jets in comparison to QCD jets even if it is not clearly discernible in the original $(x, y)$ space.

Having discussed the Fourier representation within the framework of a toy model, we now return to the Bayesian inference over $\tilde{A}_3$ with MC simulated data. The $p(\tilde{A}_3|k)$ modeling for Bayesian

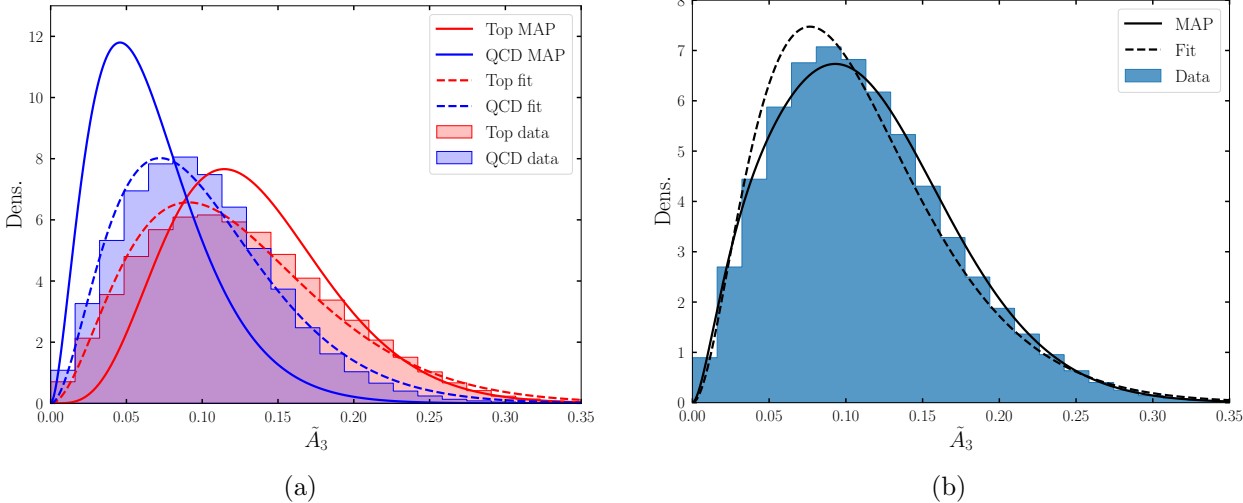

Figure 2: Left: Distribution of the variable $\tilde{A}_3$ for the QCD (top) jets in blue (red). Fitted Beta distributions for each individual class is shown in dashed lines. Our inferred MAP is shown in solid lines. Right: data without labels and weighted sum of MAP and fitted beta distributions.

inference is done as follows. Since by definition $\tilde{A}_3$ is bound between 0 and 1 we propose, as a working approximation, a probability density function corresponding to a Beta distribution for each class,

$$B(\tilde{A}_3; \alpha_k, \beta_k) = \frac{\Gamma(\alpha_k + \beta_k)}{\Gamma(\alpha_k)\Gamma(\beta_k)} \tilde{A}_3^{\alpha_k - 1}(1 - \tilde{A}_3)^{\beta_k - 1} . \tag{11}$$

which yields the appropriate domain for $\tilde{A}_3$ and is a fairly flexible functional form. Therefore, $\tilde{A}_3$ has the following total probability density given by the mixture model

$$\tilde{A}_3 \sim \pi_0 \, B(\alpha_0, \beta_0) + (1 - \pi_0) \, B(\alpha_1, \beta_1) , \tag{12}$$

where the mixing fraction $\pi_0$ is the probability of sampling a QCD jet, and consequently $(1 - \pi_0)$ corresponds to top jets. By following the general procedure described in Sec. 2 we can infer the MAP by first providing a prior. We use uniform priors for all latent variables,

$$P(\boldsymbol{\zeta}) = \begin{cases} 1, & \boldsymbol{\zeta} \in \mathcal{D} \\ 0, & \text{otherwise} \end{cases} \tag{13}$$

where $\mathcal{D}$ is a rectangular region, such that $\zeta_i \in (\zeta_i^{\min}, \zeta_i^{\max})$ for each of the model parameters. We choose $\mathcal{D}$ to be large enough to have the maximum a posteriori (MAP) away from the region boundaries.

As detailed in Sec. 2, we estimate the MAP through Stochastic Variational Inference (SVI) by using the `Pyro` package [55] in `Python3`. In Fig. 2a we plot the data distribution with true QCD (top) labels in blue (red), using a true QCD ratio of $p = 0.5$. We perform a supervised fit of a Beta distribution over each class, with the resulting distributions shown in dashed lines. Solid lines represent the distributions given by MAP estimates, and the inferred QCD probability at the MAP is $\pi_0^{\mathrm{MAP}} = 0.32$. In Fig. 2b the total data distribution is shown, where the fitted and MAP distributions are taken as the weighted sum of the corresponding Beta distributions, as in Eq. (12). Note that the MAP distribution correctly captures the data without labels, at the

cost of estimating a lower $\pi_0^{\text{MAP}}$ value. Also, for low $\tilde{A}_3$ values the top distribution given by the MAP is underestimated, which is compensated by overestimating the QCD distribution. On the other hand, the QCD distribution is underestimated at large $\tilde{A}_3$, where the top distribution is correctly reproduced. Our model succeeds at capturing some features of the class distributions and it correctly fits the data without labels. Nonetheless, given the significant overlap between both classes, it tends to infer distributions that are distant from the truth data resulting in poor tagging performance, with a top tagger based on these inferred distributions yielding an AUC of 0.62 and an accuracy of 0.55 for a sample with $p = 0.5$. This is because the mixture model is appropriately learning the total density without the need to separate explicitly into the true distributions of QCD and top jets. This is similar to the results reported in Ref. [40] where it was shown that learning the top and QCD defining features is enough for unsupervised density estimation.

These results are further backed by the posterior predictive checks. Performing the full posterior predictive check, we obtain 0.50, while for Top and QCD we obtain 0.43 and 0.32 respectively, noting that we are learning appropriately the total density distribution even if each class may be mismodelled.

A clear improvement for the Bayesian Fourier approach in this Section would be to go beyond the Beta distribution to better model the expected shape for $\tilde{A}_3$ and propose the corresponding PDF. However, even though the Fourier simplification is attractive, the approach itself has an intrinsic limitation as a tagger due to the little discrimination power one can assess from the Montecarlo data in Fig. 2a. In any case, even without a good tagger, having a better model for the $\tilde{A}_3$ distributions could provide the total fraction in the sample, as well as relevant information for the physics involved. One could also incorporate the constituent energies to the Fourier model, while preserving one-dimensionality, by using the energy fraction deposited in each $\varpi$ bin and computing the $\tilde{A}_i$ frequencies in the same fashion as above. In this case, we still observe that the Montecarlo distributions of QCD and Top data do not separate enough, and therefore the model performance is not improved.

In the next Section we develop a similar analysis as in this Section, but using the $N$-subjettiness ratio $\tau_3/\tau_2$, which shows less overlap between classes and thus it is in principle better to match density estimation to classification performance.

## 4   Mixture model for the $N$-subjettiness ratio $\tau_3/\tau_2$

A variable introduced in the literature to study jet substructure, the $N$-subjettiness [53, 60] for a given identified jet, is obtained from $N$ defined candidate subjets by computing

$$\tau_N = \frac{1}{d_0} \sum_k p_{T,k} \, \min\{(\Delta R_{1,k})^\beta, (\Delta R_{2,k})^\beta, \dots, (\Delta R_{N,k})^\beta\}, \tag{14}$$

where the sum is done over all jet constituents, $p_{T,k}$ is their transverse momenta, $\Delta R_{i,k}$ is the distance in the $(\eta, \varphi)$ plane between the $k$-th constituent and the $i$-th candidate subjet, $\beta$ is an angular weighting exponent, and $d_0$ is a normalization factor,

$$d_0 = \sum_k p_{T,k} R_0^\beta, \tag{15}$$

where $R_0$ is the original jet clustering radius. Although $N$ subjets should in principle be chosen by minimizing $\tau_N$, we follow Ref. [53, 60] and use the $N$ subjets obtained by applying the exclusive

$k_T$ clustering algorithm to the jet via `FastJet` [61]. This approximation yields a small fraction of events for which $\tau_3/\tau_2 > 1$, which is not an obstacle for our algorithm. The variable $\tau_N$ quantifies the degree to which a jet can be regarded as $N$ collimated subjets. In particular, for 3-pronged objects such as top jets, the ratio $\tau_3/\tau_2$ has shown to be an effective tagging variable using simple one-dimensional cuts [60]. To take advantage of this discriminating power, we implement the Bayesian mixture model to construct an unsupervised top tagger.

We propose a model in which $\tau_3/\tau_2$ is given by a mixture of Gamma distributions,

$$\frac{\tau_3}{\tau_2} \sim \pi_0 \, \Gamma(\alpha_0, \beta_0) + (1 - \pi_0) \, \Gamma(\alpha_1, \beta_1) \,, \tag{16}$$

where

$$\Gamma(x; \alpha, \beta) = \frac{\beta^\alpha}{\Gamma(\alpha)} x^{\alpha-1} e^{-\beta x} \,, \tag{17}$$

where $x = \tau_3/\tau_2$, and the shape and rate parameters $\alpha_k$ and $\beta_k$ are given for the QCD (top) class ($k = 0\,(1)$), and $\pi_0$ is the probability of sampling a QCD event. Ideally, the choice of $p(\tau_3/\tau_2|k)$ would be completely justified from first principles much like Softdrop multiplicity for quark/gluon tagging but this is not the case. As with $\tilde{A}_3$, simplicity and the fact that $\tau_3/\tau_2 \in [0, 1]$ would suggest that we consider a Beta distribution. However, because we are dealing with the approximate computation of $\tau_3/\tau_2$, we selected the Gamma distribution which can accommodate larger than one or close to one suboptimally computed $\tau_3/\tau_2$ values while still being fairly close to a Beta distribution. In the end, this is an arbitrary choice which is reflected in a resulting bias.

Given our mixture model we can evaluate the likelihood function of the sample data, and then construct a posterior distribution over the distribution parameters $\boldsymbol{\zeta} = (\pi_0, \alpha_0, \beta_0, \alpha_1, \beta_1)$ in an analogous manner to the one described in Sec. 2. The prior distribution we propose to be uniform in each of the inferred variables, and we infer the MAP of these parameters with SVI via the `Pyro` package; exactly as in the previous Section.

To evaluate and interpret the learned model we plot the MAP mixture components and the corresponding data distribution in Fig. 3 for a dataset with true QCD ratio $p = 0.5$. We compare the learned model both to the data and to the supervised fits to labeled data, where each Gamma distribution is fitted separately to its corresponding process. In Fig. 3a we observe that both the MAP (solid) distribution and the fitted (dashed) distributions are closer to truth level for top jets (red) than for QCD jets (blue). However, the fitted and learned QCD approximations are not identical. The MAP better captures the maximum position while the fitted distribution better reproduces the mean by sacrificing an accurate maximum in favor of accurate tails. This is caused by the differing objectives of the two approximations. The fit aims to capture the individual labelled process accurately, while the MAP aims to capture the unlabelled data distribution. In Fig. 3b we observe that the MAP captures much better the multimodality of the full data, while the fitted distributions provide a worse fit. The inadequacy of both procedures highlights the fact that for QCD the $\tau_3/\tau_2$ distribution cannot be approximated by a Gamma distribution. However, the MAP objective ensures that the MAP estimates capture the difference between QCD and top in order to explain the data more accurately, as in Sec. 3. We verify these conclusions again when performing the posterior predictive checks. Performing the full posterior predictive check, we obtain 0.49, while for Top and QCD we obtain 0.48 and 0.40 respectively, noting how we are learning appropriately the total density distribution and how the mismodelling is mainly centered on the QCD distribution.

As detailed in Sec. 2, after obtaining the MAP parameters we can construct a jet classifier by calculating the assignment probabilities of the class $\mathcal{C}_k$, with $k = 0\,(1)$ representing QCD (top)

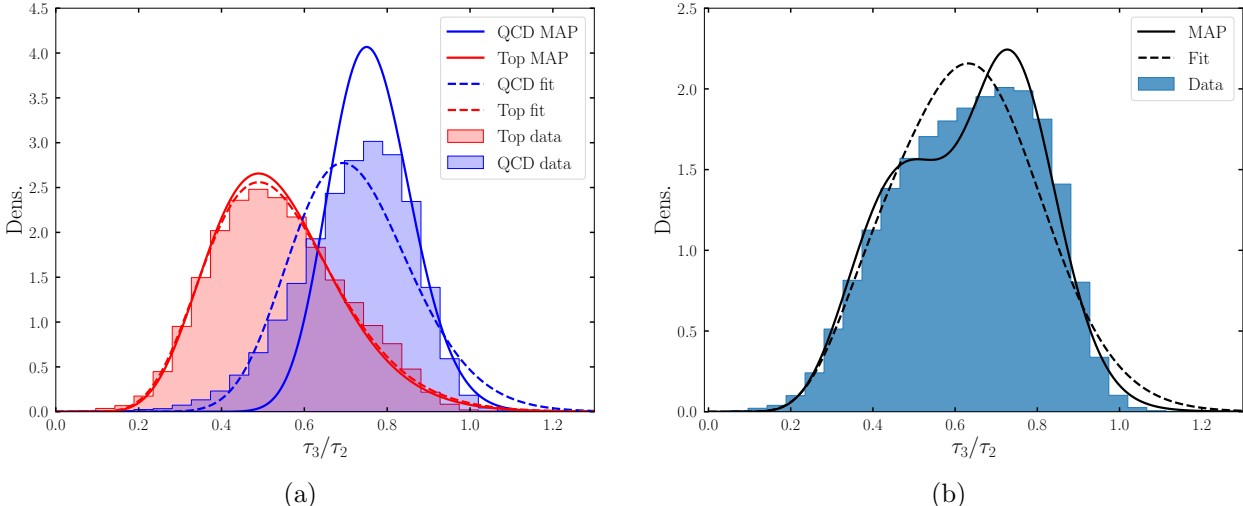

(a)          (b)

Figure 3: Distribution of the $\tau_3/\tau_2$ data with $p = 0.5$, where fitted and MAP Gamma distributions are plotted in dashed and solid lines respectively. The inferred value for the fraction in the sample is $\pi_0^{\text{MAP}} = 0.42$. Left plot shows separate QCD and top distributions, while the right plot corresponds to the total distribution. Within the allowed freedom the MAP distribution describes slightly better the data distribution, as it is what really captures the algorithm.

events,

$$P(\mathcal{C}_k | \tau_3/\tau_2, \boldsymbol{\zeta}^{\text{MAP}}) = \frac{\pi_k^{\text{MAP}} \, \Gamma(\tau_3/\tau_2; \alpha_k^{\text{MAP}}, \beta_k^{\text{MAP}})}{\sum_{i=0,1} \pi_i^{\text{MAP}} \, \Gamma(\tau_3/\tau_2; \alpha_i^{\text{MAP}}, \beta_i^{\text{MAP}})} \,. \tag{18}$$

The classifier assigns a jet to the top class if $P(\text{top}|\tau_3/\tau_2, \boldsymbol{\zeta}^{\text{MAP}}) > t$, where $t$ is a chosen probability threshold. We evaluate our tagger over the test dataset of [34] with a cut in the invariant mass of the jets, $145 \text{ GeV} < m_j < 205 \text{ GeV}$, in order to compute the usual classifier performance metrics. In Fig. 4a we show the assignment probabilities for each class given by the MAP estimates as a function of $\tau_3/\tau_2$. By varying $t$ in the interval $(0, 1)$ we can obtain the true and false positive rates (TPR and FPR respectively) and construct the ROC curve, which is plotted in Fig. 4b for the classifiers corresponding to the MAP parameters and the truth data (optimal).

From Fig. 4b, we observe that our classifier is very close to the optimal performance even if the underlying distributions do not match perfectly the true distributions. This reinforces the observation that the mixture model is learning the appropriate differences between top and QCD jets. This is also seen in Fig. 4a where the learned probabilities for each class have the appropriate behavior close to the $p(C_i) = 0.5$ in the intermediate $\tau_3/\tau_2$ region while the model is overconfident assigning very small $p(C_0)$ for small $\tau_3/\tau_2$ and underconfident with $p(C_0)$ too small for large $\tau_3/\tau_2$.

As a final test, we benchmark our model for different true fractions of QCD $p$. To compare all cases, we compute the AUC and the accuracy for each set of learned parameters over the same balanced sample of QCD and top jets unseen during inference. The results are shown in Table 1. We observe that the model performs consistently in terms of accuracy and AUC for all fractions, while the differences between $p$ and $\pi_0^{\text{MAP}}$ suggest that the model does not need to learn the exact distributions to do so. The difference between true and learned distributions is larger for larger $p$ due to the QCD distribution being the most distinct from a true Gamma distribution. In Fig. 5 we plot the true QCD fraction in the sample versus the corresponding MAP for its inferred value, which yields a satisfactory agreement of about 20% for the worst cases of large QCD fraction.

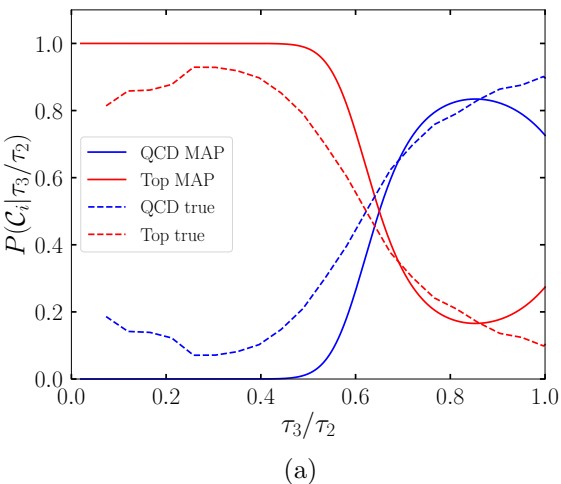
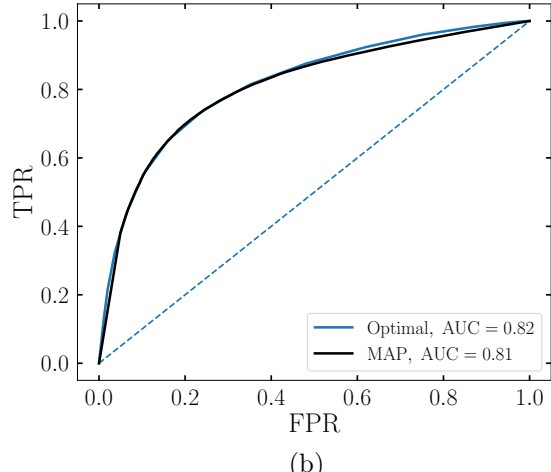

| (a) | (b) |

Figure 4: Left: Assignment probabilities for each class as a function of $\tau_3/\tau_2$, given the MAP estimates (solid lines), or given truth data (dashed). Right: ROC curves for the jet classifiers constructed with labeled truth data (blue) and MAP values (black).

| $p$ | $\pi_0^{\mathrm{MAP}}$ | Acc | AUC |
|------|------|------|------|
| 0.1 | 0.16 | 0.69 | 0.80 |
| 0.3 | 0.30 | 0.74 | 0.81 |
| 0.5 | 0.42 | 0.75 | 0.81 |
| 0.7 | 0.54 | 0.75 | 0.81 |
| 0.9 | 0.71 | 0.74 | 0.81 |

Table 1: Inferred $\pi_0^{\mathrm{MAP}}$ values and tagging performance metrics for different true mixing ratios ($p$) of QCD events.

# 5    Discussion and outlook

Along this manuscript we have explored Bayesian inference techniques through two different mixture models to unsupervisedly recognize top quark jets from QCD jets. In one model (Sec. 3), we have used as the observed variable the normalized third coefficient $\tilde{A}_3$ (Eq. 10) of the Fourier transform of the angular distributions of the tracks around the center of the jet, and proposed for its PDF a different Beta distribution for each of the classes (top and QCD). For the other model (Sec. 4), we have used as the observed variable the $N$-subjettiness ratio $\tau_3/\tau_2$, and proposed for its PDF a different Gamma distribution for each of the classes. The inference procedure has been carried out using Stochastic Variational Inference which has the virtue of being fast and reliable, at the price of obtaining the Maximum A Posteriori (MAP) instead of the full posterior.

We find that the results for the Fourier framework, which has the advantage of a pure geometrical interpretation based on the 3-pronged nature of top jets, are subpar, the main reasons being that the $\tilde{A}_3$ variable has poor discriminating power and that the modelling of its PDF through a Beta distribution is not entirely satisfactory. An alternative framework that keeps the purely geometrical interpretation would be to consider a mixture model in which the tracks of the jet are sampled either from three overlapping Normal distributions or from one Normal distribution. Such an analysis

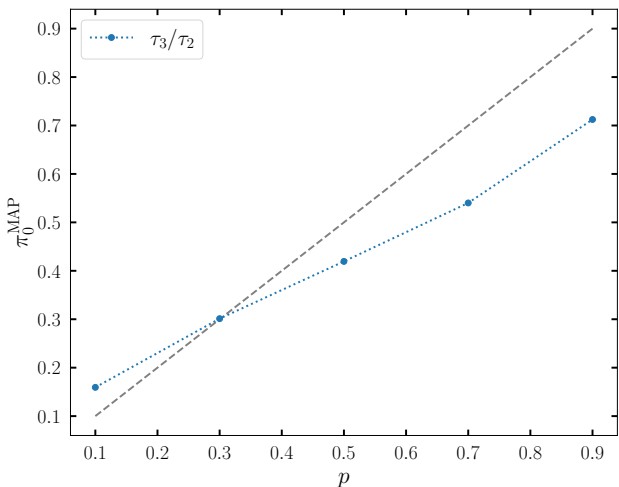

Figure 5: Learned $\pi_0^{\mathrm{MAP}}$ values as a function of true QCD mixture fraction $p$ used for inference. The agreement is slightly better for low QCD fractions, since the top distribution is better captured by the model.

seems appealing and it could have very challenging and promising approaches. For instance, the top has three final decays, but they are not all on equal conditions. Meaning that one could study and understand how this information –which is robust and from first principles– could be included in the likelihood to take better profit of it. In addition, it would be challenging to include in the likelihood that the three Normal distributions have different orientation in each event. On the other hand, the modelling for the QCD jet could also have some subtleties and going beyond a simple Normal could provide some advantages that should be studied.

The results in this work using the $N$-subjettiness ratio $\tau_3/\tau_2$ variable are considerably better than those in the Fourier framework. The PDF for the top quark class is captured well through the propose Gamma distribution. Although the PDF for $\tau_3/\tau_2$ in the QCD class is not well reproduced, still the tagger is satisfactory with an AUC $\sim 0.80 - 0.81$ and an accuracy of about $0.69 - 0.75$ for samples with top-quark fraction ranging from 10% to 90%. Moreover, the model also infers the total fraction of top-quark in the sample with very good agreement to the true value (see Fig. 5).

We find that the expected performance of the studied unsupervised taggers is below the expected performance of supervised taggers. However, the former is more robust than the latter to potential mismatches between Montecarlo simulations and real data inside the jets, since its foundations are related to robust knowledge such as for instance that the top-quark has three decay products whereas other information is inferred directly from the data itself. This kind of unsupervised taggers are more simple and robust at the price of reducing the expected performance and possible sources of bias arising from the model specification which can however be checked with Bayesian techniques such as posterior predictive checks. Another non-negligible advantage of this unsupervised tagger is that the used parameters to determine the tagging are extracted from the sample itself. That is, they are not only extracted from real data, but also they are not imported from another kinematic region.

In brief, the present work explores a class of unsupervised taggers that apply Bayesian inference on a mixture model. Our findings aim at reflecting on potential improvements in unsupervised taggers for their possible use in real analyses. In the future, one possible path is to look for more

appropriate distributions or better suited observables. For example, the energy of each constituent track could be incorporated more explicitly in the $\tilde{A}_n$. Another possible path is to increase the power of the mixture model to capture arbitrary distributions. A Normalizing Flow-inspired architecture could be developed provided the only multimodality comes from the mixture of the processes, which is the case for the observables considered in this work. This could also allow to incorporate more observables to the probabilistic model. With the current methodology, we find that the increase in bias when miss-specifying the two-dimensional distributions $p(\tilde{A}_3, \tau_3/\tau_2|k)$ offsets the increase in discrimination power.

# Acknowledgement

E.A., T.T., S.T. and A.S. thank CONICET and ANPCyT (under projects PICT 2017-2751 and PICT 2018-03682), and M.S. acknowledges support in part by the DOE grant DE-SC0011784 and NSF OAC-2103889. We all thank the *Muchachos* for the great effort poured during the development of this article.

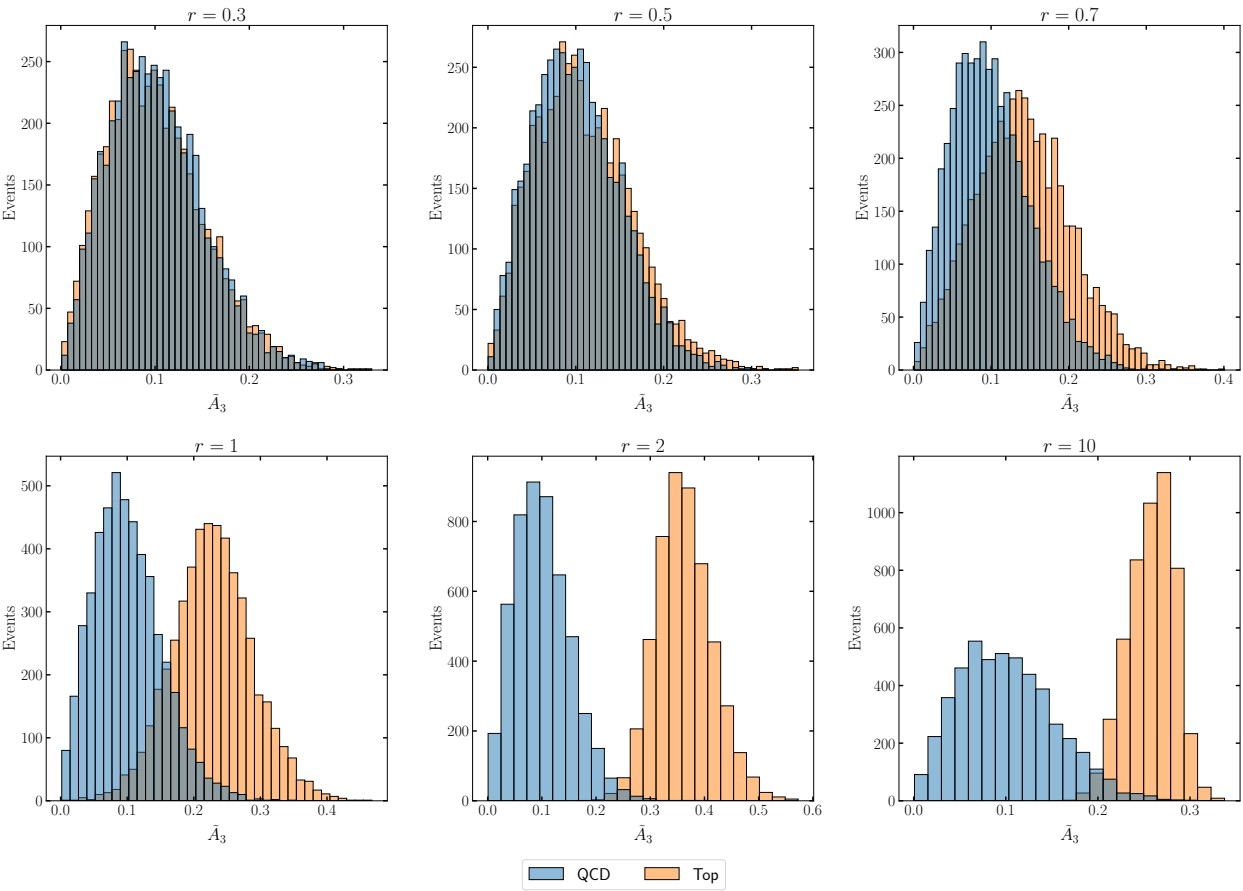

Figure 6: We show the obtained $\tilde{A}_3$ distributions for the toy QCD and top jets for different choices of the position $r$ (see the text). As $r$ increases, the differing physics between the two processes is picked up by the $\tilde{A}_3$ distributions.

# A  Toy Model

In order to test the discriminating power of the variable $\tilde{A}_3$ constructed in Sec. 3, we first build a toy model that tries to capture the basic features of the top and QCD jets. For each event in the generated data, we draw a QCD event with probability $p$, or a top event with probability $1 - p$. We represent each event by $N_i$ points, *i.e.* tracks, in the 2-dimensional plane $(x, y)$. The number of tracks $N_i$ is drawn for each event from a Poisson distribution with a given expected value $N_{\text{tr}}$. Tracks corresponding to a QCD jet are drawn from a 2-dimensional isotropic Gaussian distribution centered at the origin,

$$(x, y)_{\text{QCD}} \sim \mathcal{N}(0, \mathbf{\Sigma}_0), \quad \mathbf{\Sigma}_0 = \text{diag}(\sigma_0, \sigma_0), \tag{19}$$

while the substructure of $t \to bjj$ events is simulated as a mixture of three isotropic Gaussian distributions centered at positions $\mathbf{r}_i$, $i = 1, 2, 3$, each corresponding to a jet in the top decay,

$$(x, y)_{\text{top}} \sim p_1 \mathcal{N}(\mathbf{r}_1, \mathbf{\Sigma}_1) + p_2 \mathcal{N}(\mathbf{r}_2, \mathbf{\Sigma}_2) + p_3 \mathcal{N}(\mathbf{r}_3, \mathbf{\Sigma}_3), \quad \mathbf{\Sigma}_i = \text{diag}(\sigma_i, \sigma_i), \tag{20}$$

where $p_i$ is the probability of a particular track to correspond to one of the three jets, so $\sum_{i=1}^{3} p_i = 1$. For definiteness, in what follows we fix some of the model parameters at

$$p_1 = p_2 = p_3 = 1/3, \tag{21}$$

$$|\mathbf{r}_1| = |\mathbf{r}_2| = |\mathbf{r}_3| = r \,, \tag{22}$$

$$\text{ang}(\mathbf{r}_i, \mathbf{r}_j) = 2\pi/3 \,, \ i,j = 1,2,3 \,, \tag{23}$$

$$\sigma_i = 0.5 \,, \ i = 0,1,2,3 \,. \tag{24}$$

Therefore, with this parametrization $r$ represents how separate are the top subjets from each other. We show in Fig. 6 the $\tilde{A}_3$ distribution for different $r$ values, for both QCD and top classes. Note that the distributions become distinguishable from each other around $r \sim 0.7$, which corresponds to $r > \sigma$. On the other hand, as we increase $r$, the $\tilde{A}_3$ distribution of the top events does not reach values arbitrarily close to $\tilde{A}_3 = 1$. This is due to the fact that, as $r$ increases, so do the $l = 6, 9$ frequencies of the DFT, and since they are normalized they constrain the maximum value of $\tilde{A}_3$ for any radius. This toy model captures the essential physics and we obtain Fig. 1 by setting $r = 0.5$.

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
