# Peer review of "Exploring unsupervised top tagging using Bayesian inference"

_SciPost Physics Core_

## Round 1 · Referee Report · Anonymous (Referee 1) · 2023-2-3

Report

In this paper, authors have studied how unsupervised learning methods can be used to identify boosted top quarks from QCD jets. They claim that their approach is robust against possible Monte Carlo bias, and for this reason, this method should be helpful in

real data analysis. The concept seems interesting. However, I would like to ask a few questions before the acceptance.

Questions and comments:

1. They have discussed the possible bias in the Monte Carlo generator and cited two papers. However, a detailed discussion should be there in the paper. It would be nice if some quantitative statements could be made.

2. If we apply this method to actual data, quark jets contaminate the dataset. The quark fraction will depend on the p_T range of the jet. How does the inclusion of quark jets in the dataset change the outcome?

3. In principle, this method can also induce bias. Is it possible to estimate the bias and conclude that the bias is negligible compared to the bias of supervised methods?

4. In this paper, the authors used the nsubjettiness ratio variable tau_3/tau_2. Is it possible to

include more variables in this analysis? Supervised top-taggers use multiple

variables, and correlations among variables help to improve the accuracy.

5. For comparison, it would be good to provide the ROC using the same variable (tau3/tau_2)

using labeled dataset information.

6. In this paper, authors consider only one p_T bin. How the results vary as we change the energy bin ( say 1 TeV) is unclear.

  • validity: good
  • significance: good
  • originality: high
  • clarity: good
  • formatting: good
  • grammar: good

Author:  Santiago Tanco  on 2023-03-14  [id 3479]

(in reply to Report 1 on 2023-02-03)

We thank the reviewer for the questions and comments which we believe have improved the present work. We have made explicit all relevant changes in magenta. To answer the reviewer's comments:

They have discussed the possible bias in the Monte Carlo generator and cited two papers. However, a detailed discussion should be there in the paper. It would be nice if some quantitative statements could be made.

We have expanded the discussion on page 2 on possible biases in the Monte Carlo generators with special emphasis on how it could affect Top quark measurements. We have not provided numbers for these biases as these would require an in-detail study and would not be straight-forwardly comparable to any metric derived from our method.

If we apply this method to actual data, quark jets contaminate the dataset. The quark fraction will depend on the p_T range of the jet. How does the inclusion of quark jets in the dataset change the outcome?

The reviewer is correct in that the quark composition depends on the $p_T$. In this case, the quark jet contribution is already contained in the QCD sample and thus would not change anything for the narrow $p_T$ bin which was used to generate the dataset. We have modified the text on page 2 to make explicit the fact that QCD contains both gluons and quarks. The $p_{T}$ dependence of the observables can be taken into account by binning in $p_T$ and training different models for each bin or by incorporating $p_T$ as another variable and modifying the probabilistic model accordingly. This is feasible if a good guess for the $p_T$ dependence of the relevant variables is known.

In principle, this method can also induce bias. Is it possible to estimate the bias and conclude that the bias is negligible compared to the bias of supervised methods?

The reviewer is correct in that the method also induces bias. In our case, the bias resides in the choice of the class-dependent PDFs. In a semi-supervised framework, simulations can be considered to check whether the obtained probability distributions match the desired probability distributions. This is what we did for N-subjettiness and found that the choice of Gamma induces the wrong shape for the QCD distribution. In a fully unsupervised case, other metrics such as the total Kullback-Leibler divergence or more powerful checks such as a posterior predictive check could be implemented. This series of checks are part of bayesian model building and, in opposition to traditional supervised analysis, are easily interpretable. We do not claim that our method as presented here is currently better than a supervised analysis as the proposed functional forms do bias the results, but we claim that it is a way forward for more robust taggers that are simulation independent. We have clarified this in the main text and added posterior predictive checks to emphasize that, although biased, the model is still accomplishing the density estimation task it is designed to do. The motivation and definition of this check can be found on pages 5-6, while the results are given in pages 10 and 11 for the different mixture models.

In this paper, the authors used the nsubjettiness ratio variable tau_3/tau_2. Is it possible to include more variables in this analysis? Supervised top-taggers use multiple variables, and correlations among variables help to improve the accuracy.

The reviewer is correct in that a multidimensional input is a key feature of the powerful top taggers. In our case, including multiple variables is equivalent to specifying a multidimensional distribution with a particular correlation structure. If this can be done either by choosing independent variables or by an accurate modelling of the multidimensional distribution, multiple variables can be included.

For comparison, it would be good to provide the ROC using the same variable (tau3/tau_2) using labeled dataset information.

We have changed Fig. 4b in order to show the ROC calculated using the labeled dataset, instead of the fitted probability density functions. Nonetheless, these ROC curves are very similar and this change is hardly noticeable in the plot.

In this paper, authors consider only one p_T bin. How the results vary as we change the energy bin ( say 1 TeV) is unclear.

As we mentioned in the previous response, the reviewer is correct in that the results are $p_T$ dependent. Although we have not explored it, at higher $p_T$ the taggers will have a degradation in performance as Top and QCD jets become more alike and the observables themselves lose discrimination power. We have added a footnote clarifying the $p_T$ dependence of the taggers when we introduce the "Top Quark Tagging Reference Dataset" in Section 2.

---

## Round 1 · Referee Report · Tilman Plehn (Referee 2) · 2023-2-6

Strengths

- Resilience of subjet taggers is one of the key questions, and independent of the question if this tagger will eventually be used, it is important to test and publish ideas which might eventually help.

Weaknesses

- Concerning the general case: I am not at all convinced that top taggers need to be trained on MC, because there is data to train on, and then MC to check. For top-tagging this strategy is self-inflicted damage by the experiments.

- Looking at the method: zeta appears between Eq.(1) and Eq.(3), and I am not sure I get the logic of k vs zeta, and classification vs regression. Please stick to one problem in all formulas.
- Can you put your Bayesian method into ML-context, like the conditional flow (cINN) or BayesFlow inference. Generative models for Bayesian inference are popping up everywhere these days.
- Related to the ML methods, how about extending this method to more than one dimension?
- Sorry, to understand the Fourier philosophy, what happens beyond angles, for instance the constituent energies?
- I am not sure I understand the QCD toy model, should that not be a peak at small angles? Maybe I am not getting it, it that is the case, I apologize in advance.
- Please introduce the SVI a little more carefully, since it is key to the method.

- Moving to the results: for the N-subjettiness model, why a Gamma distribution?
- Most importantly, should that method not work for more than one observable? We know that single-observable taggers are not good. Except for maybe Softdrop, why not use that one? Or jet mass correlated with N-subjettiness?
- Why to the authors believe their method does not introduce a bias? What about the choice of shape functions? Maybe try two and see what happens for one of their cases?

- A few comments on the references: why cite the Higgs discovery in a top tagging paper?
- I agree that top physics is a great future program, but please give some motivation.
- In the introduction, please give credit to BDRS for traditional subjet tagging, and please cite the original HEPTopTagger paper - replacing the later [14]?
- For the subjet taggers, please mention the original jet images paper.
- Please cite the ML-taggers compared in Ref.[18], to help the original authors of the different ML top taggers. It's not that many...

- Finally, in the conclusions I do not understand the sentence `it could have very challenging and promising approaches' on p.10 means what?

Report

As mentioned above, the paper is not super-brilliant, but it is more than interesting enough to publish in SciPost Core. My comments are mostly concerned with the presentation, but maybe the authors could think about and discuss the physics questions as well.

Requested changes

See the weaknesses, they can call be fixed easily.

  • validity: high
  • significance: high
  • originality: high
  • clarity: high
  • formatting: excellent
  • grammar: perfect

Author:  Santiago Tanco  on 2023-03-14  [id 3480]

(in reply to Report 2 by Tilman Plehn on 2023-02-06)

We thank Tilman Plehn for the very detailed review with questions and comments which we believe have improved the present work. We have made explicit all relevant changes in magenta. To answer the different comments:

Concerning the general case: I am not at all convinced that top taggers need to be trained on MC, because there is data to train on, and then MC to check. For top-tagging this strategy is self-inflicted damage by the experiments.

We agree. We consider the Top Quark Tagger dataset only for benchmarking but the aim of this method is to be applied directly on data.

Looking at the method: zeta appears between Eq.(1) and Eq.(3), and I am not sure I get the logic of k vs zeta, and classification vs regression. Please stick to one problem in all formulas.

The notation was selected to separate between class-membership and model parameters for $p(x|param)$. Here $k$ is the class index and $\zeta$ the model parameters which are random variables in the Bayesian framework. $\zeta$ includes both the binomial probability parameter $\pi_{k}$ and the class-dependent parameters $\zeta_{k}$. We build a classifier out of the probability of a given class assignment $C_k$ given the data and the parameters $p(C_k|x,\zeta)$. There is no regression in this set-up, only density estimation for classification.

Can you put your Bayesian method into ML-context, like the conditional flow (cINN) or BayesFlow inference. Generative models for Bayesian inference are popping up everywhere these days.

We have added a bit of context on page 3, comparing our method to other generative processes in the literature. Because the literature is so vast, we have mentioned only a subset of the relevant HEP papers to show other approaches and their differences with ours. Mainly, we show how we are aiming to capture the multimodality of the data in a way that can be matched to true physical processes and which relies on simulations as little as possible. We have not included a BayesFlow reference because we believe the other references included also deal with learning surrogate models with Invertible Networks and are directly applied to High Energy Physics.

Related to the ML methods, how about extending this method to more than one dimension?

To extend the method to more dimensions we would need to specify a multidimensional distribution with a particular correlation structure. As we mention in Section 5, we found that the additional bias can offset the benefit of including more variables. However, as with selecting better one-dimensional distributions, there are possible ways forward by going to more sophisticated probabilistic models.

Sorry, to understand the Fourier philosophy, what happens beyond angles, for instance the constituent energies?

The observation is very suitable. In fact, one could consider how much energy is deposited in each {\it slice} and extract the information using these values to compute the Discrete Fourier Transform. We have computed this alternative and we encounter the same problem: the true distributions for signal and background do not separate enough. The overall performance is not better when using energies instead of counting tracks. We have added this information to the manuscript on page 10.

I am not sure I understand the QCD toy model, should that not be a peak at small angles? Maybe I am not getting it, it that is the case, I apologize in advance.

If we understand this question correctly, the Referee observes that in fact in the toy model QCD is modeled as a Normal distribution around the center of the jet and hence it populates the small angles nearby the center. In the Fourier analysis we take slices whose corners are in the center of the jet. Therefore in this case, with all tracks near the center, we expect all of the slices to get fairly equally populated in tracks, since it does not affect whether the tracks are near or far from the center, but rather their distribution in the angle sweeping the (theta, phi) plane. Therefore, this distribution should not provide a special activation for the $\tilde A_3$ component.

Please introduce the SVI a little more carefully, since it is key to the method.

We have added a more detailed description of SVI on pages 4 and 5.

Moving to the results: for the N-subjettiness model, why a Gamma distribution?

Ideally, the shape function choice would be completely justified from first principles much like Softdrop multiplicity for quark/gluon tagging. However, when this is not the case, simplicity is perhaps the guiding principle. In this sense, we first considered a Beta distribution because this ratio of N-subjettiness observables should be a random number in the [0,1] range. However, because we are dealing with the approximate computation of N-subjettiness, we selected the Gamma distribution which can accommodate larger than 1 or close to 1 suboptimally computed N-subjettiness values while still being fairly close to a Beta distribution. In the end, this is an arbitrary choice which is reflected in the resulting bias which could be bettered. We have added a sentence clarifying this methodology on page 11.

Most importantly, should that method not work for more than one observable? We know that single-observable taggers are not good. Except for maybe Softdrop, why not use that one? Or jet mass correlated with N-subjettiness?

We did not consider Softdrop as an initial variable because we found that the N-subjettiness distribution and its differences between Top and QCD could be chosen less arbitrarily. However, our method could easily accomodate SoftDrop once a shape function choice is made, specially because we found our choice of shape functions for N-subjettiness to be imperfect. In regards to combining observables, and as discussed in the multidimensionality response, we did not incorporate multidimensional outputs because of the difficulty on modelling multidimensional correlations.

Why to the authors believe their method does not introduce a bias? What about the choice of shape functions? Maybe try two and see what happens for one of their cases?

Our method does indeed introduce bias in the choice of the shape functions. This is why we cannot appropriately recover the QCD distribution. However, this bias can be verified explicitly within a Bayesian framework. We have added one such method, the posterior predictive check, in the main text to show how our method may be biased for N-Subjettiness but is still able to recover the approximate total density distribution. A general description of the method was added on pages 5 and 6, while the results for the different mixture models are given on pages 10 and 11. As discussed in the multidimensionality response, incorporating more general shape functions with appropriate diagnostics can reduce the bias while still being more robust than MC taggers.

A few comments on the references: why cite the Higgs discovery in a top tagging paper?

We included these references in the first sentence of the Introduction as a mere example to illustrate LHC success. However, it is true that in the context of the paper they may seem a bit off-topic, so we decided to remove them.

I agree that top physics is a great future program, but please give some motivation.

We have added some motivation on top physics in Section 1.

In the introduction, please give credit to BDRS for traditional subjet tagging, and please cite the original HEPTopTagger paper - replacing the later [14]?

We modified the text to give credit to BDRS and we have cited the original HEPTopTagger papers. We didn't remove [14] as it is still a valuable reference.

For the subjet taggers, please mention the original jet images paper.

We have added a reference to the original jet-images tagger.

Please cite the ML-taggers compared in Ref.[18], to help the original authors of the different ML top taggers. It's not that many...

We have added references to the original ML top taggers cited in [18].

Finally, in the conclusions I do not understand the sentence `it could have very challenging and promising approaches' on p.10 means what?

We agree with the Referee that further discussion is needed after this sentence. We have added a few sentences which help to visualize the appealing and scope of the proposed idea.

---

## Round 2 · Referee Report · Tilman Plehn · 2023-3-16

Report
Thank you to the authors for going through all my comments and complaints, I think the paper is ready to be published.

---

## Round 2 · Referee Report · Anonymous · 2023-4-14

Report
I am happy with the changes made by authors. This paper should be published.

---

## Round 2 · List of Changes

Following the reports of both reviewers, we have made changes to the manuscript that are explicitly shown in magenta

You are currently on this page

---

## Editorial Decision

editorial_decision: